# A Modified Recommended Food Score Is Inversely Associated with High Blood Pressure in Korean Adults

**DOI:** 10.3390/nu12113479

**Published:** 2020-11-12

**Authors:** Kyuyoung Han, Yoon Jung Yang, Hyesook Kim, Oran Kwon

**Affiliations:** 1Department of Nutritional Science and Food Management, Ewha Womans University, 52, Ewhayeodae-gil, Seodaemun-gu, Seoul 03760, Korea; kyuhan1017@gmail.com; 2System Health & Engineering Major in Graduate School, Ewha Womans University, 52, Ewhayeodae-gil, Seodaemun-gu, Seoul 03760, Korea; 3Department of Food and Nutrition, Dongduk Women’s University, 60, Hwarang-ro 13-gil, Seongbuk-gu, Seoul 02748, Korea; Yoonjung.yang@gmail.com

**Keywords:** high blood pressure, hypertension, Recommended Food Score (RFS), modified Recommended Food Score (mRFS), Dietary Approaches to Stop Hypertension (DASH) diet, KNHANES

## Abstract

Hypertension is associated with an increase in cardiovascular disease and mortality. The interplay between dietary intake—especially sodium intake—and high blood pressure highlights the importance of understanding the role of eating patterns on cardiometabolic risk factors. This study investigates the relationship between a modified version of the Recommended Food Score (RFS) and hypertension in 8389 adults aged 19–64 years from the Korea National Health and Nutrition Examination Survey 2013–2015. A dish-based, semi-quantitative, 112-item food frequency questionnaire was used to assess dietary intakes. Modified RFS (mRFS) is based on the reported consumption of foods recommended in the Dietary Approaches to Stop Hypertension (DASH) diet modified for Korean foods. High blood pressure included hypertension and prehypertension, also known as stage 1 hypertension. Men and women with the highest quintile of mRFS had a 27.2% (OR: 0.728, 95% CI: 0.545–0.971, *p*-trend = 0.0289) and 32.9% (OR: 0.671, 95% CI: 0.519–0.867, *p*-trend = 0.0087) lower prevalence of high blood pressure than those with the lowest quintile of mRFS, respectively. Our finding suggests that a higher mRFS may be associated with a lower prevalence of high blood pressure among the Korean adult population.

## 1. Introduction

Hypertension, or high blood pressure, is one of the most common diseases worldwide. It is a risk factor for cardiovascular diseases and all-cause mortality [1,2,3]. Although there are several approaches to lower blood pressure (e.g., pharmacological treatment), experts recommend lifestyle modifications as the initial treatment strategy [4]. In 2020, the International Society of Hypertension (ISH) developed and reported worldwide practice guidelines for the management of hypertension, including salt reduction, healthy diet, healthy drinks, moderation of alcohol consumption, weight reduction, smoking cessation, regular physical activity, stress reduction, mindfulness, and reduced exposure to air pollution and cold temperature [5]. Among these lifestyle modifications, the Dietary Approaches to Stop Hypertension (DASH) diet is a dietary pattern promoted by the USA-based National Heart, Lung, and Blood Institute (NHLBI) of the National Institutes of Health (NIH) to prevent and control hypertension [6].

The DASH diet emphasizes the foods (fruits, vegetables, whole grains, lean protein, and low-fat dairy) high in blood pressure-deflating nutrients, like potassium, calcium, protein, and fiber. It also discourages foods that are high in saturated fat, such as fatty meats, full-fat dairy foods, and tropical oils, as well as sugar-sweetened beverages and sweets. The DASH diet score, developed by Fung et al. [7], is an index to determine adherence to the DASH diet. Several studies have shown that strict compliance with the DASH diet is necessary to achieve any of its benefits [8,9,10].

However, calculating the DASH diet score is not simple. Therefore, a rapid and simple measure of adherence to the DASH diet is needed to help physicians deal effectively with the prevention and management of hypertension or high blood pressure among the general population. The Recommended Food Score (RFS), developed by Kant et al. [11], is a simple indicator of overall diet quality that assigns a score of 1 or 0, depending on whether or not the food that is recommended in the dietary guidelines is consumed more than once a week. Many studies have reported the relationship between RFS and several diseases, such as hypertension [12], metabolic syndrome [13], cancer [14], and mortality [15].

In this study, we calculated the modified RFS (mRFS), which is simpler than the existing DASH diet score, by applying the RFS scoring system. We then examined whether mRFS is related to high blood pressure in the Korean adult population.

## 2. Materials and Methods

### 2.1. Study Population

This study is based on data from the first three years (2013–2015) of the Korea National Health and Nutrition Examination Survey (KNHANES) VI conducted by the Korea Disease Control and Prevention Agency (KDCA), formerly known as the Korea Centers for Disease Control and Prevention (KCDC). KNHANES uses multistage, stratified, probability-clustered sampling to select the primary sampling units—households belonging to non-institutional residents in South Korea. The survey consists of a health interview survey, a health assessment survey, and a nutrition survey. KNHANES was approved by the KCDC Institutional Review Board (IRB) until 2014, and has been proceeding without deliberation according to the bioethics law since 2015 (2013-07CON-03-4C, 2013-12EXP-03-5C). Written informed consent was obtained from all participants. More information about the survey can be found on the KNHANES website (http://knhanes.cdc.go.kr).

A total of 22,948 (9380 men, 11,692 women) participants enrolled in the survey. We excluded those aged under 19 and over 64 years (*n =* 9423), pregnant and lactating women, and those with missing values for mRFS (*n* = 2998). We subsequently excluded those with implausible energy intakes of <500 or >8000 kcal/day (*n* = 53), those with missing values on energy intake, and those without systolic blood pressure (SBP) and diastolic blood pressure (DBP) measurements (*n* = 582). Finally, we excluded participants diagnosed with hypertension (*n* = 1260) and those prescribed a blood pressure regulator (*n* = 6). Participants with other chronic diseases and participants on various medications were included. As a result, a final sample of 8389 participants (3327 men, 5062 women) formed our study population.

### 2.2. General Characteristics

We obtained the demographic and socio-economic characteristics of the participants, including age, body mass index (BMI), household income, smoking status, alcohol consumption, menopause status in women only (pre-menopause, post-menopause), and blood pressure (SBP, DBP) values. BMI was calculated as weight (kg)/height (m)^2^. Household income was classified into quartiles of low, middle-low, middle-high, and high. Smoking status was categorized as a non-smoker, ex-smoker, and current smoker. Alcohol consumption was characterized as never, once a month, 2–4 times a week, and ≥4 times a week. Regular exercise was classified into “yes” or “no”. To confirm if the participants exercised regularly, we used walking practice rate, moderate physical activity practice rate, and vigorous physical activity practice rate as variables. If any of these variables were included, we classified participants as “yes” in a regular exercise category. There are differences in survey categories among national health and nutrition surveys. The walking practice rate is the percentage of practicing walking for at least 30 min for at least five days a week for the last week, and the rate of practicing moderate physical activity for the last week was at least 30 min of moderate physical activity for five days a week. In this study, we analyzed the data from 2014–2015 using the index of the fraction of practicing physical activity for at least 10 min at a time for at least three days a week as high-intensity physical activity.

### 2.3. High Blood Pressure

The definition of hypertension is an SBP ≥ 140 mmHg, or DBP  ≥  90 mmHg, or self-reported use of antihypertensive medication [16]. Prehypertension is defined as an SBP ≥ 120 and ≤ 140 mmHg and DBP ≥ 80 and ≤ 90 mmHg [17]. In this study, we defined high blood pressure as including hypertension and prehypertension, also known as stage 1 hypertension.

### 2.4. mRFS

The dish-based, semi-quantitative food frequency questionnaire (FFQ) conducted by KNHANES was used to analyze the participants’ dietary intake. Consumption frequency was recorded as “almost never” to “three times per day”, and participants chose a serving size option from three categories: A half of, equal to, or 1.5 to 2.0 times a standard serving size. In this study, we constructed an mRFS by utilizing the RFS based on the reported frequency of consumption of recommended and unrecommended foods in the DASH diet and modified for Korean foods. The RFS is a validated index of overall diet quality [11] and considered useful for predicting biomarkers of dietary intake, chronic disease, and mortality [18,19,20]. The RFS scoring system assigns 1-point each time a recommended food is consumed more than once a week, and 1-point when a non-recommended food is consumed less than once a week. The original RFS food groups are based on fruits, vegetables, grains, meat, and dairy [11]. We reclassified the food groups into nine categories: Whole grains, nuts, and legumes, fruits/fruit juices, vegetables, low-fat milk, fish, processed meat, sugar-sweetened beverages, and sodium-rich food. Table 1 lists all foods or food groups in these nine categories.

The DASH diet score should be calculated separately for the total sodium intake, but the mRFS tool was developed to simply apply this process to food. Among the 119 FFQ items, we included 13 items classified as sodium-rich food. These foods include fermented fish products [21], Korean stew [22], kimchi [23], pickled vegetables [24], and noodles [25], which have been reported as salty foods, due to their high salt content.

A score of 1 was assigned to participants who consumed any of the recommended foods (whole grains, nuts, and legumes, fruits/fruit juices, vegetables, low-fat milk, and fish) more than once a week, and a score of 0, otherwise. In addition, participants who consumed any of the non-recommended foods (processed meat, sugar-sweetened beverages, and sodium-rich food) more than once a week received a score of 0, or a score of 1 if the frequency of consumption was less than once a week. The total mRFS score range was 0–66.

### 2.5. Statistical Analysis

All statistical analyses were performed using SAS software version 9.4 (SAS Institute, Cary, NC, USA). Due to the complex sampling design of the KNHANES study, the relevant primary sampling units, stratification, and sample weights were considered in our analysis. The participants were categorized into quintiles according to their mRFS by gender. For descriptive statistics, the continuous variables are expressed as weighted means and standard deviations, and categorical variables are expressed as number and weighted percentages, using the SURVEYMEANS and SURVEYFREQ procedures, respectively. For the trend test in SURVEYREG models, the quintile group of mRFS were treated as continuous variables assigned with the median value within each category. To estimate the odds ratios (ORs) and 95% confidence intervals (CIs) for the risk of high blood pressure across mRFS quintiles, SURVEYLOGISTIC analysis was performed with the 1st quintile (Q1) set as a reference. Model 1 was adjusted for the covariates sex, age, and menopause status (women only). Model 2 was further adjusted for the covariates BMI (kg/m^2^), household income (low, middle-low, middle-high, high), smoking status (non-smoker, ex-smoker, current smoker), drinking status (never, once a month, 2–4 times a month, 2–3 times a week, ≥4 times a week), regular exercise (yes/no), and energy intake. All reported probability tests were considered bi-directional. The level of significance was set at 5%.

## 3. Results

### 3.1. General Characteristics

Table 2 characterizes the participants by the mRFS quintiles. Compared to those in a lower mRFS quintile group, both men and women participants in the higher mRFS quintile group tend to be older, less likely to be a current smoker and heavy drinker, have a higher energy intake, and exercise regularly (*p*-trend < 0.0001). In addition, the mean score of each recommended food group in mRFS (whole grains, nuts, and legumes, fruits/fruit juices, vegetables, low-fat milk, and fish) tended to increase across the higher mRFS quintile groups (*p*-trend < 0.0001). The mean score of each unrecommended food group in mRFS, except for the sodium-rich food group (red and processed meat and sugar-sweetened beverages), also tended to increase across the higher mRFS quintile groups both in men and women (*p*-trend < 0.0001).

Table 3 shows the mean and SD of SBP and DBP in each mRFS quintile after adjusting for potential covariates. Model 1 was adjusted for sex, age, and menopause status (women only). Model 2 was adjusted for sex, age, BMI, household income, smoking, alcohol drinking, regular exercise, energy intake, and menopause status (women only). The mean SBP in Model 1 tended to decrease across the higher mRFS quintile group in both men and women (men: *p*-trend = 0.0314, women: *p*-trend < 0.0001). In Model 2, the mean SBP tended to decrease across the higher mRFS quintile group only in women (*p*-trend = 0.0003). Moreover, compared to those in a lower mRFS quintile group, participants in the higher mRFS quintile group had a lower mean DBP in Model 1 in both men and women (men: *p*-trend = 0.0250, women: *p*-trend < 0.0001). The mean DBP in Model 2 tended to decrease across the higher mRFS quintile group both in men and women (men: *p*-trend = 0.0425, women: *p*-trend = 0.0011).

### 3.2. Relationship between mRFS and Risk of Prevalence of High Blood Pressure

Table 4 represents the OR with 95% CI for high blood pressure based on the mRFS. When the mRFS was analyzed as a continuous variable, there was a significant inverse relationship between the mRFS and high blood pressure in Model 1, and except for women, in Model 2. In Model 1, as the mRFS increases by 1 point, the prevalence of high blood pressure was lowered by 6.6% (OR: 0.934, 95% CI: 0.879–0.993) in men and 6.9% (OR: 0.931, 95% CI: 0.883–0.982) in women. These results indicate that as the mRFS score increases, the risk of high blood pressure decreases. For the total participants, the risk of high blood pressure in Model 1 was 27.7% lower in Q5 compared to Q1 (OR: 0.723, 95% CI: 0.594–0.880, *p*-trend = 0.0032). In Model 2, the risk of prevalence of high blood pressure was 26.7% lower in Q5 compared to Q1 (OR: 0.733, 95% CI: 0.593–0.906, *p*-trend = 0.0075). For men, the risk of prevalence of high blood pressure in Model 1 and Model 2 were 27.2% and 30.9% lower in Q5 compared to Q1 (OR: 0.728, 95% CI: 0.545–0.971, *p*-trend = 0.0388 and OR: 0.691, 95% CI: 0.501–0.953, *p*-trend = 0.0220) in order. Finally, for women, the risk of high blood pressure in Model 1 and Model 2 were 32.9% and 25.0% in Q5 compared to Q1 (OR: 0.671, 95% CI: 0.519–0.867, *p*-trend = 0.0045 and OR: 0.750, 95% CI: 0.569–0.989, *p*-trend = 0.0946), respectively.

## 4. Discussion

In this study, we calculated the mRFS for Korean adult men and women by utilizing the RFS modified for Korean foods based on foods emphasized in the DASH diet. We confirmed the presence of an inverse relationship between the mRFS and the risk of high blood pressure.

Hypertension is affected by various genetic and environmental factors, including diet—especially sodium intake—and eating habits. The DASH diet is recognized for its efficacy in lowering hypertension [26]. Developed by Fung et al. [7], the DASH diet score reflects the adherence to the DASH diet, which is the dietary pattern recommended by the NHLBI for preventing and controlling hypertension [27,28,29]. However, the procedure to calculate the DASH diet score is complicated [11]. It requires calculating the total intake of the food groups included in the DASH diet and dividing the scores into quintiles. The total amount of sodium must be calculated separately, and the sodium score divided into quintiles. Finally, each food group’s scores are summed and added to the sodium score for the total score. High blood pressure is a common condition. Therefore, a simpler and easier method than the DASH diet score is needed to gauge the DASH diet adherence and has public health applicability for the general population.

To calculate the RFS based on foods recommended in the DASH diet in a simple manner, we used FFQ data collected by KNHANES and constructed an mRFS based on the RFS scoring system modified for Korean foods. We assigned 1 point depending on whether a food recommended in the DASH diet was consumed more than once a week. After analyzing the association between the mRFS and blood pressure, we observed that the higher the mRFS quintile, the lower the blood pressure (*p*-trend < 0.0001; Table 3). Accordingly, men and women in mRFS Q5 had, respectively, a 30.9% and 25.0% lower risk of prevalence of high blood pressure compared to those in mRFS Q1. These results suggest the possibility that mRFS may be associated with a lower prevalence of high blood pressure. In addition, in men corresponding to Q4, as well as mRFS Q5, the risk of prevalence of high blood pressure was 27.3% lower than that of men in Q1. Such findings indicate that by following a DASH-style diet to a certain degree, the risk of having high blood pressure may be lowered. However, an in-depth study is needed to examine its validity and confirm whether it has applicability for the general public.

For this study, we excluded participants diagnosed with hypertension and prescribed a blood pressure regulator. We included participants with prehypertension, also known as stage 1 hypertension, because the DASH diet aims to prevent high blood pressure. In addition, we analyzed the mRFS for both genders, specifically because blood pressure can be affected by hormones. Ong et al. [30] showed that there is a gender difference in blood pressure control. Moreover, Staessen et al. [31] suggested that ovarian deficiency potentiates the age-related increase in SBP in women. As the results in Table 2 demonstrate, we found that the mRFS based on the recommended foods in the DASH diet (whole grains, nuts, and legumes, fruits/fruit juice, vegetables, low-fat milk, and fish) was higher in Q5 than Q1 for men, women, and the total participants. By contrast, the mRFS based on the unrecommended foods (red and processed meat, sugar-sweetened beverages, and sodium-rich food) was lowest in Q5 for men, women, and the total participants (Table 2). In other words, the participants with higher mRFS consumed the recommended foods more and the unrecommended foods less than those who had a lower mRFS. In particular, the results of the inverse relationship between mRFS and high blood pressure in this study are thought to be related to the relatively higher intake of fruits and vegetables in the Q5 group.

To calculate the mRFS, we used the data collected by a semi-quantitative FFQ developed for Koreans and administered by the KNHANES VI. The FFQ is useful for ascertaining long-term dietary patterns of individuals and populations and has been found to contribute some important information about diet–disease associations [32]. The most important function of the FFQ, which represents a useful food intake survey method to identify the dietary factors of chronic diseases, is to properly reflect the dietary habits of the group or individual surveyed [33].

In calculating the mRFS, we focused on nine food groups (whole grains, nuts, and legumes, fruits/fruit juices, vegetables, low-fat milk, fish, red and processed meat, sugar-sweetened beverages, sodium-rich food). To evaluate the effect of sodium intake, several salty foods were placed among the foods in the FFQ (Table 1). However, *ssamjang* and red chili-pepper paste with vinegar are major traditional Korean fermented food types [34]. Large amounts of sodium are added for the preservation of traditional Korean foods [35]. Despite the high-sodium content, the traditional Korean food kimchi contains vitamin C, vitamin K, dietary fiber, and an abundance of lactic acid bacteria, such as *Lactobacillus plantarum*, *Lactobacillus brevis*, and *Pediococcus cerevisiae* [36]. Another high-sodium food is bean paste stew/rich soybean paste stew, made from soybean paste. Soybean paste (doenjang) is another traditional Korean fermented food with high-sodium content. Attempts to develop salt-reduced fermented soybean products, such as doenjang, reported negative sensory attributes and decreased consumer acceptance ratings [37]. Although most of the items included in the sodium-rich food group can increase blood pressure, it is necessary to consider both the fact that they are fermented foods and the beneficial effects of cooking them with other vegetables. Therefore, this aspect warrants further discussion and may be a limitation of this study.

Another limitation of this study is its cross-sectional design, which means that inferences about associations between the mRFS and high blood pressure cannot be confirmed. According to Plumer et al. [38], this is due to the evaluation method—the FFQ, which is generally based on the local food supplies and national surveys. Hence, the results are not applicable to other societies. Moreover, mRFS is simply an index that evaluates the quality of diet associated with high blood pressure, and does not reflect total energy intake or energy needs. It should also be noted that without any external validation data, we cannot confirm that the mRFS is measuring what it purports to do; hence, further validation of this index is required. Nonetheless, as far as we know, this is the first study to calculate the RFS based on foods recommended in the DASH diet in a simple manner and identify whether this is related to high blood pressure in the Korean adult population. This study has the potential to inform developers of dietary guidelines about the role of diet quality or dietary patterns on the health of adults.

High blood pressure is a common chronic health condition in adults and a risk factor for cardiovascular disease, brain vascular disease, and kidney disease [39]. Thus, it is important to prevent high blood pressure and recognize one’s health status in advance. Understanding the relationship between diet quality and high blood pressure allows for proper intervention to prevent hypertension and associated illnesses by improving health. The mRFS may be a useful and simple tool to recognize whether one’s diet is at the stage of risk for high blood pressure. This scoring instrument can also be applied easily to many clinical studies because of its simplicity.

## 5. Conclusions

We found inverse associations between the mRFS and the risk of high blood pressure for both men and women in the Korean adult population. These results suggest that a higher mRFS may be associated with a lower prevalence of high blood pressure. Further studies, with relatively larger sample sizes and a prospective or interventional design, are needed to improve our knowledge about the association between the mRFS and high blood pressure in the future.

## Figures and Tables

**Table 1 nutrients-12-03479-t001:** Foods or food groups in the mRFS.

Food Subgroups	Number of Foods Included	Foods Included
Whole grain	1	Multigrain rice
Nuts and legumes	5	Tofu/braised tofu/fried tofu, boiled bean, peanut, soy milk, marron
Fruits/Fruit juice	13	Strawberry, oriental melon, watermelon, peach, grape, apple, pear, dried persimmon/persimmon, tangerine, banana, orange, kiwifruit, fruit juice
Vegetables	15	Green-bean sprouts/bean sprouts, spinach, balloon flower, pumpkin, other vegetables, cucumber, daikon, vegetable salad, chives/green onion, broccoli/cabbage, garlic, lotus root/burdock, mushroom, leafy vegetables/green chili, cherry tomato/tomato
Low-fat milk	1	Low-fat milk
Fish	3	Mackerel pike/mackerel, croaker/hairtail, stir-fried anchovy/anchovies
Red and processed meat	12	Stock soup of bone and stew meat/beef-bone soup, pork back-bone stew, beef stew/hot spicy meat stew/radish soup, pork/grilled pork belly, boiled pork, pork bulgogi/steamed pork ribs/stir-fried spicy pork/grilled spareribs, pork cutlet/sweet and sour pork, roast beef, beef bulgogi, sausage stew, ham, Korean sausage
Sugar-sweetened beverage	3	Carbonated beverage (cola, sprite, carbonated fruit beverage), powder made of mixed grains/sikhye (sweet rice drink), coffee with sugar
Sodium-rich food	13	Salted crab, salted shrimp/salted squid/salted clam, ssamjang (Korean red pepper paste, soybean paste, mixed paste)/red chili-pepper paste with vinegar, Korean instant noodle (ramyeon)/Korean instant cup noodle, noodles/chopped noodles/udon, black bean paste noodles/spicy seafood noodles, bean paste stew/rich soybean paste stew, tofu stew/soft tofu stew, kimchi stew/stir-fried kimchi, pollack stew/spicy seafood stew, cabbage kimchi, other types of kimchi/fresh kimchi dressed with garlic and chili powder, pickled vegetable/pickled cucumber

**Table 2 nutrients-12-03479-t002:** General characteristics of the participants by quintile (Q1–Q5) of the mRFS.

Total	Q1	Q2	Q3	Q4	Q5	*p*-Trend
*n*	1394	2180	1495	1600	1720	
Age, years	36.9 ± 11.3	41.1 ± 12.2	42.4 ± 12.2	44.2 ± 12.1	45.7 ± 11.7	<0.0001
BMI, kg/m^2^	23.3 ± 3.8	23.6 ± 3.5	23.3 ± 3.4	23.5 ± 3.2	23.4 ± 3.2	0.2282
Household income						
Low	140 (9.6)	226 (9.9)	130 (7.7)	120 (7.1)	102 (5.4)	<0.0001
Middle-Low	421 (31.3)	558 (25.2)	366 (23.9)	338 (21.2)	334 (17.8)	
Middle-High	465 (32.5)	704 (32.7)	494 (34.7)	502 (31.6)	509 (31.3)	
High	364 (26.6)	684 (32.2)	499 (33.7)	631 (40.1)	766 (45.5)	
Missing, *n*	*4*	*8*	*6*	*9*	*9*	
Smoking status						
Non-smoker	779 (52.7)	1257 (52.0)	916 (57.9)	1027 (61.7)	1149 (64.0)	<0.0001
Former smoker	196 (16.0)	317 (17.3)	254 (20.1)	255 (18.0)	261 (17.4)	
Current smoker	362 (31.3)	536 (30.7)	262 (22.0)	244 (20.3)	232 (18.6)	
Missing *n*	*57*	*70*	*63*	*74*	*78*	
Drinking						
Never	203 (12.4)	397 (15.9)	285 (17.1)	353 (19.6)	439 (23.0)	<0.0001
Once a month	426 (31.0)	649 (29.7)	492 (32.1)	515 (33.7)	567 (33.3)	
2–4 times a month	371 (29.4)	584 (30.3)	371 (29.0)	386 (26.9)	375 (25.5)	
2–3 times a week	242 (19.8)	363 (18.0)	217 (16.5)	209 (15.3)	212 (15.0)	
≥4 times a week	95 (7.4)	118 (6.1)	67 (5.3)	64 (4.5)	49 (3.2)	
Missing, *n*	*57*	*69*	*63*	*73*	*78*	
Regular exercise						
No	627 (46.3)	951 (43.1)	596 (39.7)	561 (36.0)	569 (32.5)	<0.0001
Yes	679 (53.7)	1104 (56.9)	795 (60.3)	912 (64.0)	1026 (67.5)	
Missing, *n*	*88*	*125*	*104*	*127*	*125*	
Menopause status						
Pre-menopause	1316 (96.6)	1978 (93.8)	1336 (93.1)	1383 (90.1)	1405 (83.9)	<0.0001
Post-menopause	78 (3.4)	202 (6.2)	159 (6.9)	217 (9.9)	315 (13.1)	
Energy intake, kcal/day	2124.1 ± 845.6	1933.1 ± 790.8	1983.0 ± 735.5	2060.4 ± 760.2	2243.5 ± 855.8	0.9746
Carbohydrate, % total energy	59.2 ± 13.4	61.0 ± 13.9	61.9 ± 13.0	62.8 ± 13.1	63.1 ± 12.4	<0.0001
Protein, % total energy	13.9 ± 4.4	13.7 ± 3.9	13.9 ± 3.9	13.8 ± 3.8	14.4 ± 3.7	0.0428
Fat, % total energy	21.5 ± 8.8	20.2 ± 9.1	19.8 ± 8.6	19.8 ± 8.7	20.1 ± 8.5	<0.0001
mRFS, range	12–26	27–29	30–31	32–34	35–55	
mRFS, score out of 66	24.8 ± 1.9	28.2 ± 1.3	30.7 ± 1.1	33.1 ± 1.3	38.4 ± 3.3	<0.0001
Whole grains, score out of 1	0.6 ± 0.5	0.8 ± 0.4	0.9 ± 0.3	0.9 ± 0.3	0.9 ± 0.2	<0.0001
Nuts and legumes, score out of 5	0.3 ± 0.5	0.4 ± 0.6	0.5 ± 0.7	0.8 ± 0.8	1.3 ± 1.0	<0.0001
Fruits/Fruit juices, score out of 13	0.9 ± 1.1	1.4 ± 1.4	2.1 ± 1.6	3.0 ± 1.8	4.6 ± 2.4	<0.0001
Vegetables, score out of 15	1.7 ± 2.0	2.1 ± 2.1	3.3 ± 2.4	4.4 ± 2.6	7.3 ± 3.0	<0.0001
Low-fat milk, score out of 1	0.1 ± 0.3	0.1 ± 0.3	0.2 ± 0.4	0.2 ± 0.4	0.3 ± 0.5	<0.0001
Fish, score out of 3	0.3 ± 0.6	0.4 ± 0.7	0.7 ± 0.7	0.9 ± 0.8	1.3 ± 1.0	<0.0001
Red and processed meat, score out of 12	10.4 ± 1.9	11.1 ± 1.3	11.1 ± 1.3	11.1 ± 1.4	11.1 ± 1.4	<0.0001
Sugar-sweetened beverage, score out of 3	1.7 ± 0.7	1.9 ± 0.7	2.0 ± 0.7	2.1 ± 0.7	2.1 ± 0.7	<0.0001
Sodium-rich food, score out of 13	8.0 ± 1.9	8.9 ± 1.9	8.9 ± 2.0	8.7 ± 2.0	8.4 ± 1.9	<0.0001
Men						
*n*	537	917	631	640	602	
Age, years	37.0 ± 11.4	40.4 ± 12.3	42.3 ± 12.5	43.3 ± 12.6	44.4 ± 12.3	0.0004
BMI, kg/m^2^	24.4 ± 3.5	24.4 ± 3.5	24.0 ± 3.2	24.3 ± 3.1	24.4 ± 3.3	0.4405
Household income						
Low	46 (8.2)	89 (9.7)	49 (7.4)	39 (6.0)	32 (5.0)	<0.0001
Middle-Low	151 (30.4)	218 (23.8)	163 (25.0)	134 (21.7)	101 (15.3)	
Middle-High	191 (34.0)	308 (33.6)	210 (35.2)	209 (32.5)	182 (32.4)	
High	149 (27.4)	299 (32.9)	208 (32.4)	256 (39.8)	281 (47.3)	
Missing *n*	-	3	1	2	6	
Smoking status						
Non-smoker	107 (22.5)	199 (23.6)	171 (30.8)	171 (31.6)	149 (29.6)	<0.0001
Former smoker	122 (23.6)	243 (26.3)	194 (31.7)	216 (31.9)	223 (34.4)	
Current smoker	276 (53.9)	445 (50.1)	227 (37.5)	209 (36.5)	194 (36.0)	
Missing, *n*	32	30	39	44	36	
Drinking						
Never	41 (7.2)	91 (9.6)	66 (10.8)	85 (13.3)	86 (14.2)	<0.0001
Once a month	100 (21.6)	190 (22.3)	140 (23.3)	131 (23.3)	139 (25.5)	
2–4 times a month	166 (32.9)	298 (34.9)	196 (35.9)	189 (32.6)	169 (30.9)	
2–3 times a week	133 (26.1)	216 (23.5)	136 (21.7)	139 (23.0)	140 (24.3)	
≥4 times a week	65 (12.2)	92 (9.7)	54 (8.3)	52 (7.8)	32 (5.1)	
Missing, *n*	32	30	39	44	36	
Regular exercise						
No	213 (42.4)	361 (39.8)	237 (37.9)	190 (32.0)	184 (30.7)	<0.0001
Yes	279 (57.6)	497 (60.2)	339 (62.1)	382 (68.0)	364 (69.3)	
Missing *n*	45	59	55	68	56	
Menopause status						
Pre-menopause	-	-	-	-	-	-
Post-menopause	-	-	-	-	-	
Energy intake, kcal/day	2542.8 ± 847.8	2307.1 ± 836.7	2277.5 ± 793.0	2384.4 ± 804.4	2698.9 ± 978.2	<0.0001
Carbohydrate, % total energy	56.3 ± 13.6	58.2 ± 14.4	59.8 ± 13.5	59.3 ± 13.7	60.0 ± 13.7	<0.0001
Protein, % total energy	14.3 ± 4.2	13.7 ± 3.9	14.0 ± 3.8	13.8 ± 3.8	14.6 ± 4.0	0.1780
Fat, % total energy	21.5 ± 8.6	20.1 ± 8.9	19.5 ± 8.2	20.4 ± 8.4	20.0 ± 8.5	0.0273
mRFS, range	12–25	26–28	29–30	31–33	34–51	
mRFS, score out of 66	23.6 ± 1.8	27.1 ± 0.8	29.5 ± 0.5	31.9 ± 0.8	36.9 ± 3.1	<0.0001
Whole grains, score out of 1	0.6 ± 0.5	0.8 ± 0.4	0.8 ± 0.4	0.9 ± 0.3	0.9 ± 0.3	<0.0001
Nuts and legumes, score out of 5	0.3 ± 0.6	0.4 ± 0.7	0.5 ± 0.8	0.8 ± 0.8	1.4 ± 1.1	<0.0001
Fruits/Fruit juices, score out of 13	0.8 ± 1.1	1.1 ± 1.2	1.6 ± 1.6	2.4 ± 1.7	3.9 ± 2.4	<0.0001
Vegetables, score out of 15	1.8 ± 2.2	2.1 ± 2.2	3.2 ± 2.6	4.4 ± 2.7	7.5 ± 3.2	<0.0001
Low-fat milk, score out of 1	0.1 ± 0.3	0.1 ± 0.3	0.1 ± 0.4	0.2 ± 0.4	0.3 ± 0.4	<0.0001
Fish, score out of 3	1.3 ± 1.8	1.5 ± 1.9	1.9 ± 2.0	2.4 ± 2.7	3.6 ± 3.1	<0.0001
Red and processed meat, score out of 12	9.9 ± 2.1	11.0 ± 1.5	11.0 ± 1.5	11.0 ± 1.4	10.7 ± 1.6	<0.0001
Sugar-sweetened beverage, score out of 3	1.5 ± 0.7	1.7 ± 0.7	1.8 ± 0.7	1.9 ± 0.7	1.9 ± 0.8	<0.0001
Sodium-rich food, score out of 13	7.4 ± 2.0	8.6 ± 1.9	8.7 ± 2.1	8.6 ± 2.0	8.0 ± 2.0	0.0860
Women						
*n*	857	1263	864	960	1118	
Age, years	36.9 ± 11.1	41.6 ± 12.2	42.4 ± 12.0	44.7 ± 11.8	46.3 ± 11.4	<0.0001
BMI, kg/m^2^	22.6 ± 3.8	23.0 ± 3.4	22.8 ± 3.5	22.9 ± 3.2	22.9 ± 3.1	0.1085
Household income						
Low	94 (10.9)	137 (10.2)	81 (8.2)	81 (8.3)	70 (5.8)	<0.0001
Middle-Low	270 (32.1)	340 (26.8)	203 (22.5)	204 (20.7)	233 (19.9)	
Middle-High	274 (31.1)	396 (31.7)	284 (34.1)	293 (30.6)	327 (30.4)	
High	215 (25.9)	385 (31.3)	291 (35.2)	375 (40.4)	485 (43.9)	
Missing *n*	*4*	*5*	*5*	*7*	*63*	
Smoking status						
Non-smoker	672 (79.6)	1058 (85.2)	745 (88.4)	856 (92.2)	1000 (92.5)	<0.0001
Former smoker	74 (9.3)	74 (6.8)	60 (7.0)	39 (4.0)	38 (3.4)	
Current smoker	86 (11.1)	91 (8.0)	35 (4.6)	35 (3.8)	38 (4.1)	
Missing *n*	25	40	24	30	42	
Drinking						
Never	162 (17.1)	306 (23.2)	219 (24.2)	268 (26.0)	353 (30.2)	<0.0001
Once a month	326 (39.4)	459 (38.5)	352 (41.9)	384 (44.3)	428 (39.9)	
2–4 times a month	205 (26.2)	286 (25.0)	175 (21.3)	197 (21.2)	206 (21.0)	
2–3 times a week	109 (14.1)	147 (11.5)	81 (10.7)	70 (7.4)	72 (7.2)	
≥4 times a week	30 (3.2)	26 (1.8)	13 (1.9)	12 (1.1)	17 (1.7)	
Missing *n*	25	39	24	29	42	
Regular exercise						
No	414 (49.8)	590 (46.7)	359 (41.9)	371 (40.0)	385 (34.1)	<0.0001
Yes	400 (50.2)	607 (53.3)	456 (58.1)	530 (60.0)	662 (65.9)	
Missing, *n*	43	66	13	9	71	
Menopause status						
Pre-menopause	779 (93.6)	1061 (86.6)	705 (85.1)	743 (79.6)	803 (75.8)	<0.0001
Post-menopause	78 (6.4)	202 (13.4)	159 (14.9)	217 (20.4)	315 (24.2)	
Energy intake, kcal/day	1861.8 ± 731.2	1661.4 ± 629.2	1767.9 ± 606.4	1844.4 ± 644.8	1998.3 ± 663.6	<0.0001
Carbohydrate, % total energy	61.0 ± 12.9	63.0 ± 13.1	63.5 ± 12.3	65.2 ± 12.1	64.8 ± 11.4	0.0011
Protein, % total energy	13.7 ± 4.4	13.7 ± 3.9	13.9 ± 3.9	13.8 ± 3.8	14.3 ± 3.6	<0.0001
Fat, % total energy	21.5 ± 9.0	20.2 ± 9.3	20.1 ± 8.7	19.4 ± 8.9	20.2 ± 8.6	0.0994
mRFS, range	15–27	28–30	31–32	33–35	36–55	
mRFS, score out of 66	25.6 ± 1.6	29.1 ± 0.8	31.5 ± 0.5	34.0 ± 0.8	39.2 ± 3.1	<0.0001
Whole grains, score out of 1	0.7 ± 0.5	0.9 ± 0.4	0.9 ± 0.3	0.9 ± 0.2	0.9 ± 0.2	<0.0001
Nuts and legumes, score out of 5	0.3 ± 0.5	0.4 ± 0.6	0.5 ± 0.7	0.8 ± 0.8	1.3 ± 1.0	<0.0001
Fruits/Fruit juices, score out of 13	0.9 ± 1.1	1.6 ± 1.4	2.5 ± 1.6	3.4 ± 1.8	5.0 ± 2.4	<0.0001
Vegetables, score out of 15	1.6 ± 1.8	2.2 ± 2.0	3.3 ± 2.3	4.6 ± 2.5	7.1 ± 2.9	<0.0001
Low-fat milk, score out of 1	0.1 ± 0.3	0.1 ± 0.3	0.2 ± 0.4	0.2 ± 0.4	0.3 ± 0.5	<0.0001
Fish, score out of 3	0.3 ± 0.6	0.4 ± 0.7	0.7 ± 0.7	1.0 ± 0.9	1.3 ± 0.9	<0.0001
Red and processed meat, score out of 12	10.7 ± 1.6	11.3 ± 1.2	11.2 ± 1.2	11.1 ± 1.3	11.2 ± 1.2	<0.0001
Sugar-sweetened beverage, score out of 3	1.8 ± 0.7	2.1 ± 0.7	2.1 ± 0.7	2.2 ± 0.7	2.3 ± 0.7	<0.0001
Sodium-rich food, score out of 13	8.2 ± 1.9	9.1 ± 1.9	9.0 ± 1.9	8.7 ± 1.9	8.7 ± 1.8	0.4220

Data are expressed as mean ± SD or frequency (%). BMI, body mass index; mRFS, modified Recommended Food Score; SBP, systolic blood pressure; DBP, diastolic blood pressure; Q, quintile. For the trend test in SURVEYREG models, the quintile group of mRFS were treated as continuous variables assigned with the median value within each category.

**Table 3 nutrients-12-03479-t003:** Means and standard deviations of SBP and DBP by quintiles (Q1–Q5) of mRFS.

		Q1	Q2	Q3	Q4	Q5	*p-Trend*
Systolic Blood Pressure							
Total							
	*n*	1394	2180	1495	1600	1720	
	Model 1	113.6 ± 0.4	113.0 ± 0.3	112.7 ± 0.4	112.4 ± 0.4	111.3 ± 0.4	<0.0001
	Model 2	113.3 ± 0.4	112.8 ± 0.3	112.9 ± 0.4	112.3 ± 0.4	111.5 ± 0.4	0.0004
Men							
	*n*	537	917	631	640	602	
	Model 1	117.7 ± 0.6	117.1 ± 0.4	116.9 ± 0.5	116.0 ± 0.5	116.1 ± 0.6	0.0314
	Model 2	117.0 ± 0.7	116.7 ± 0.5	117.1 ± 0.5	115.7 ± 0.6	115.8 ± 0.6	0.0813
Women							
	*n*	857	1263	864	960	1118	
	Model 1	111.2 ± 0.5	110.3 ± 0.4	110.1 ± 0.5	110.3 ± 0.5	107.9 ± 0.5	<0.0001
	Model 2	111.1 ± 0.5	110.3 ± 0.4	110.2 ± 0.5	110.5 ± 0.5	108.7 ± 0.5	0.0003
Diastolic Blood Pressure							
Total							
	*n*	1394	2180	1495	1600	1720	
	Model 1	75.4 ± 0.3	75.0 ± 0.2	74.6 ± 0.3	74.2 ± 0.3	73.8 ± 0.3	<0.0001
	Model 2	75.2 ± 0.3	75.0 ± 0.2	74.9 ± 0.3	74.2 ± 0.3	74.0 ± 0.3	0.0004
Men							
	*n*	537	917	631	640	602	
	Model 1	79.1 ± 0.5	78.5 ± 0.4	78.1 ± 0.4	77.5 ± 0.4	77.8 ± 0.5	0.0250
	Model 2	78.5 ± 0.5	78.0 ± 0.4	78.2 ± 0.4	76.9 ± 0.4	77.4 ± 0.5	0.0425
Women							
	*n*	857	1263	864	960	1118	
	Model 1	72.9 ± 0.3	72.7 ± 0.3	72.4 ± 0.4	72.2 ± 0.3	71.1 ± 0.3	<0.0001
	Model 2	73.1 ± 0.4	73.0 ± 0.3	72.8 ± 0.4	72.7 ± 0.4	71.8 ± 0.3	0.0011

Data are expressed as mean ± SD. Model 1: Adjusted by sex, age, and menopause status (women only). Model 2: Adjusted by sex, age, menopause status (women only), BMI, household income, smoking status, drinking status, regular exercise, and energy intake. BMI, body mass index; mRFS, modified Recommended Food Score; Q, quintile. For the trend test in SURVEYREG models, the quintile group of mRFS were treated as continuous variables assigned with the median value within each category.

**Table 4 nutrients-12-03479-t004:** Odds ratios and 95% confidence intervals for quintiles (Q1–Q5) of high blood pressure based on the mRFS.

	Continuous	Q1	Q2	Q3	Q4	Q5	*p*-Trend
Total							
*n*	8389	1394	2180	1495	1600	1720	
Model 1	0.939 (0.901, 0.979)	1.000 (reference)	0.797 (0.669, 0.951)	0.836 (0.688, 1.015)	0.772 (0.644, 0.926)	0.723 (0.594, 0.880)	0.0032
Model 2	0.943 (0.902, 0.987)	1.000 (reference)	0.824 (0.683, 0.995)	0.912 (0.742, 1.120)	0.816 (0.673, 0.989)	0.733 (0.593, 0.906)	0.0075
*Men*							
*n*	3327	537	917	631	640	602	
Model 1	0.934 (0.879, 0.993)	1.000 (reference)	0.767 (0.598, 0.985)	0.832 (0.635, 1.088)	0.691 (0.532, 0.897)	0.728 (0.545, 0.971)	0.0388
Model 2	0.927 (0.866, 0.991)	1.000 (reference)	0.798 (0.610, 1.044)	0.972 (0.724, 1.305)	0.727 (0.551, 0.959)	0.691 (0.501, 0.953)	0.0220
*Women*							
*n*	5062	857	1263	864	960	1118	857
Model 1	0.931 (0.883, 0.982)	1.000 (reference)	0.806 (0.625, 1.040)	0.811 (0.612, 1.075)	0.852 (0.660, 1.098)	0.671 (0.519, 0.867)	0.0045
Model 2	0.956 (0.901, 1.015)	1.000 (reference)	0.837 (0.642, 1.091)	0.840 (0.628, 1.125)	0.932 (0.713, 1.218)	0.750 (0.569, 0.989)	0.0946

Values are expressed as ORs and 95% CI. Model 1: Adjusted by sex, age, and menopause status (women only). Model 2: Adjusted by sex, age, menopause status (women only), BMI, household income, smoking status, drinking status, regular exercise, and energy intake. OR, odds ratio; CI, confidence interval; BMI, body mass index; mRFS, modified Recommended Food Score; Q, quintile. For the trend test in SURVEYREG models, the quintile group of mRFS were treated as continuous variables assigned with the median value within each category.

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
