# Peer review of "A Modified Recommended Food Score Is Inversely Associated with High Blood Pressure in Korean Adults"

_nutrients, 2020, doi:10.3390/nu12113479_

Round 1

Reviewer 1 Report

The authors aimed to evaluate the relationship between dietary quality and blood pressure in a Korean population in a cross-sectional study. It is of merit to assess such relationships, as they can be different from those observed in western countries due to the different dietary patterns. The authors used a scoring system (mRFS) based on dish-based food frequency questionnaire to reflect participants' dietary conformance to the DASH diet. Below are my comments: 

  1. Limitation of the RFS:

The RFS does not seem to take into account total energy intake, or energy needs for different individuals. For the same MRFS for each food group, a person with low energy needs and a person with high energy needs could have very different diets.

Line 105: It seems that the mRFS have additional shortcomings as an indicator of diet quality. For example, the food groups were divided without regards to how they were prepared, Sodium from salt itself was not assessed, fruit juices were not further divided to 100% juice or juice with added sugar. In addition, please clarify if the list of foods in table 1 was exhaustive, or simply serves as an example.

Even though the authors stated this a limitation (Line 229, line 191), statements praising the mRFS are also presented in the manuscript (Line 190, 240), thus presenting a contradictory message. Due to these shortcomings, the impact of this study is diminished.

  1. The paragraph starting line 71: Please clarify if 1) participants with other chronic diseases were included; 2) participants on various medications were included.
  2. One of the missing discussion points is that only those in Q4 or Q5 were had significantly lower blood pressure based on Model 2. The authors should consider discussing how their diets differed. For example, for participants in Q5, It seems that they consume a high amount of vegetables, with a score of 19, somewhat high fruit intake, but the intake of other food groups did not differ substantially, even though significantly. This presents two interesting messages: 1) there is room for improvement, even for the Q5 group. 2) The public message for such results, is that only if you follow a DASH style diet to a certain degree, you may be at lower risk of having high blood pressure.
  3. Based on the characteristics of the participants in Q1, it seems that they could be young adults, who do not live a healthy lifestyle, yet, their blood pressure may be still normal because of the younger age, as a result, the results may be confounded when using them as the reference group. Have the authors explored the study population by age groups? How different age groups’ mRFS differed, as well as their blood pressure? How about only evaluating middle aged adults? 
  4. Table 1.

1) The authors should consider adding labels for significantly different group comparisons. Currently, with only a P-value, it is not clear which groups were different without referring to the text.

2) some of the data in table 1 did not have an SD.

3) for some of the categorical variables, it seems that the n does not add up to the total n of that quintile. For example, n=1394 for Q1 for both genders together, but the number of participants with household income data was 1390. Were there missing data? Please clarify how missing data were treated?

  1. The authors may inadvertently submitted table 3 as table 2.
  2. The authors discussed the use of the mRFS as an indicator of DASH diet adherence (Lines 190, 239). However, the study was not designed to assess such claim, and the results of the study do not support the statements.
  3. In paragraph starting with line 211, the author discussed the potential conflicting impacts of Ssamjang and soybean paste due to their unique nutrient composition. Have the authors considered that these items may also have different impacts because of how they were used? For example, soybean paste maybe often cooked together with other vegetables.
  4. For the purpose of transparency, please briefly state whether statistical analysis model assumptions were checked and validated.

Author Response

Reviewer 1

The authors aimed to evaluate the relationship between dietary quality and blood pressure in a Korean population in a cross-sectional study. It is of merit to assess such relationships, as they can be different from those observed in western countries due to the different dietary patterns. The authors used a scoring system (mRFS) based on dish-based food frequency questionnaire to reflect participants' dietary conformance to the DASH diet. Below are my comments:

→ Authors: We sincerely appreciate the reviewer's insightful and constructive comments and suggestions. Please see our detailed responses below.

  1. Limitation of the RFS:

The RFS does not seem to take into account total energy intake, or energy needs for different individuals. For the same mRFS for each food group, a person with low energy needs and a person with high energy needs could have very different diets.

→ Authors: Thank you for your comment. As the statement highlights, the mRFS is simply an index to evaluate diet quality on high blood pressure, and not to reflect total energy intake or energy needs. We have added this information to the limitations of the study (Lines 259–260). However, we excluded subjects with a food energy intake <500 kcal/day or >8,000 kcal/day during the subject selection process (Lines 73–74). In addition, when examine the relationship between mRFS and high blood pressure, we analyzed the relationship between them by adjusting the total energy intake (Table 3 and Table 4).

Line 105: It seems that the mRFS have additional shortcomings as an indicator of diet quality. For example, the food groups were divided without regards to how they were prepared, Sodium from salt itself was not assessed, fruit juices were not further divided to 100% juice or juice with added sugar. In addition, please clarify if the list of foods in table 1 was exhaustive, or simply serves as an example.

Even though the authors stated this a limitation (Line 229, line 191), statements praising the mRFS are also presented in the manuscript (Line 190, 240), thus presenting a contradictory message. Due to these shortcomings, the impact of this study is diminished.

→ Authors: We appreciate the reviewer's critical comment. We agree with the limitations of mRFS stated in the comment. We used the dish-based, semi-quantitative FFQ, designed and developed for the Korea National Health and Nutrition Examination Survey (KNHANES), to identify foods or dishes associated with high blood pressure. However, because we used the food list of the FFQ, it was difficult to consider how to cook the food items. Furthermore, the FFQ does not mention whether fruit juice contains added sugar, so this could not be considered. The DASH diet score should be calculated separately for the total sodium intake, but the mRFS tool was developed to simply apply this process to food. Among the 119 FFQ items, we included 13 items classified as sodium-rich food. These foods include fermented fish products [21], Korean stew [22], kimchi [23], pickled vegetable [24], and noodles [25], which have been reported as salty foods due to their high salt content. We have added this information to the Method section (Lines 117–121). In addition, Table 1 is not a simple example but rather lists all the foods or dish names included in the nine categories (Lines 113–116).

  1. The paragraph starting line 71: Please clarify if 1) participants with other chronic diseases were included; 2) participants on various medications were included.

→ Authors: Participants with chronic diseases, for example, dyslipidemia, myocardial infarction, diabetes, and cancer, were all included. We only excluded participants already diagnosed with hypertension and prescribed a blood pressure regulator. We added these details to the Study Population section (Lines 77–78).

  1. One of the missing discussion points is that only those in Q4 or Q5 were had significantly lower blood pressure based on Model 2. The authors should consider discussing how their diets differed. For example, for participants in Q5. It seems that they consume a high amount of vegetables, with a score of 19, somewhat high fruit intake, but the intake of other food groups did not differ substantially, even though significantly. This presents two interesting messages: 1) there is room for improvement, even for the Q5 group. 2) The public message for such results, is that only if you follow a DASH style diet to a certain degree, you may be at lower risk of having high blood pressure.

→ Authors: Thank you for the valuable comment. Based on your comments, we have added this information to the Discussion section (Lines 214–218).

  1. Based on the characteristics of the participants in Q1, it seems that they could be young adults, who do not live a healthy lifestyle, yet, their blood pressure may be still normal because of the younger age, as a result, the results may be confounded when using them as the reference group. Have the authors explored the study population by age groups? How different age groups’ mRFS differed, as well as their blood pressure? How about only evaluating middle aged adults?

→ Authors: Thank you for the valuable comment. We agree with the opinion given. However, a recent study revealed that the pre-hypertension level among Korean individuals was the highest at 24.5% in those in their 40s [a], and adults aged 30 to 40 showed the highest increase rate. In other words, it is emphasized that those in their 30s and 40s are the early hypertensive patients and are an important age group for preventing hypertension. Therefore, in this study, we included all adults aged 19–64 years. Moreover, studies on factors related to hypertension are mostly conducted in adults [b–d].

[a] Korea Centers for Disease Control and Prevention. The fifth Korea national health statistics National Health and Nutrition Examination Survey (KNHANES V). Seoul: Korea Centers for Disease Control and Prevention; 2018. p. 1–727.

[b] Hasan, M., Sutradhar, I., Akter, T., Das Gupta, R., Joshi, H., Haider, M.R., Sarker, M. Prevalence and determinants of hypertension among adult population in Nepal: Data from Nepal Demographic and Health Survey 2016. PLoS One 2018, 13, e0198028.

[c] Anteneh, Z.A., Yalew, W.A., Abitew, D. Prevalence and correlation of hypertension among adult population in Bahir Dar city, northwest Ethiopia: a community based cross-sectional study. Int. J. Gen. Med. 2015, 8, 175–185.

[d] Burt, V.L., Cutler, J.A., Higgins, M., Horan, M.J., Labarthe, D., Whelton, P., Brown, C., Roccella, E.J. Trends in the prevalence, awareness, treatment and control of hypertension in the adult US population. Data from the Health Examination Surveys, 1960 to 1991. Hypertension 1995, 26, 60–69.

  1. Table 1.

1) The authors should consider adding labels for significantly different group comparisons. Currently, with only a P-value, it is not clear which groups were different without referring to the text

→ Authors: The mRFS is a score with continuous values. However, in this study, we divided the mRFS into quintile groups. In this way, variables that were originally continuous are classified into tertiles, quartiles, or quintiles according to their distribution in many studies. When comparing the mean of certain variables by this group, statistical significance is often shown by the p-trend [e–g]. In other words, the analysis is performed under the assumption that there is a linear relationship between mRFS and a certain variable (factor). For this reason, we did not modify the original p-trend results in Table 2.

[e] Eng, J.Y., Moy, F.M., Bulgiba, A., Rampal, S Dose–response relationship between western diet and being overweight among teachers in Malaysia. Nutrients 2020, 12, 3092.

[f] Joo, N.-S., Dawson‐Hughes, B., Kim, Y-.S., Oh, K., Yeum, K.-J. Impact of calcium and vitamin d insufficiencies on serum parathyroid hormone and bone mineral density: analysis of the Fourth and Fifth Korea National Health and Nutrition Examination Survey (KNHANES IV-3, 2009 and KNHANES V-1, 2010). JBMR 2013, 28, 764–770.

[g] Park, S.-H., Lee, K.-S., Park, H.-Y. Dietary carbohydrate intake is associated with cardiovascular disease risk in Korean: Analysis of the third Korea National Health and Nutrition Examination Survey (KNHANES III). Int. J. Cardiol. 2010, 139, 234–240.

2) Some of the data in table 1 did not have an SD.

→ Authors: Thank you for these comments. Accordingly, we revised Table 2 by adding the missing SD.

3) for some of the categorical variables, it seems that the n does not add up to the total n of that quintile. For example, n=1394 for Q1 for both genders together, but the number of participants with household income data was 1390. Were there missing data? Please clarify how missing data were treated?

→ Authors: We sincerely appreciate this important comment. The missing data were not included in the original table. We added the missing data to the table in the revised manuscript—these missing data were not included in the relevance analysis (Table 2).

  1. The authors may inadvertently submit table 3 as table 2.

→ Authors: We inadvertently copied the results from Table 4 into Table 3. We apologize for this oversight. Table 3 has been properly re-edited (Table 3).

  1. The authors discussed the use of the mRFS as an indicator of DASH diet adherence (Lines 190, 239). However, the study was not designed to assess such claim, and the results of the study do not support the statements.

→ Authors: Thank you for your valuable comments. In the process of applying the foods emphasized in the DASH diet using the dish-based FFQ consisting of foods consumed by Koreans, several limitations have arisen. As mentioned in the comment, there are limitations to the use of mRFS as an indicator of DASH diet adherence, so we revised the sentence in a way that clearly describes the results of this study rather than using this expression (Lines 56–58, 206, 212–218, 260–262, 271–272).

  1. In paragraph starting with line 211, the author discussed the potential conflicting impacts of Ssamjang and soybean paste due to their unique nutrient composition. Have the authors considered that these items may also have different impacts because of how they were used? For example, soybean paste maybe often cooked together with other vegetables.

→ Authors: Thank you for these valuable comments. For ssamjang and red chili-pepper paste with vinegar, the frequency of intake is questioned as a separate item from the question asked about the intake frequency of various types of vegetables. This is thought to be because Koreans often consume ssamjang or red chili-pepper paste with vinegar as a dip when they eat vegetables. Although foods such as bean paste stew/rich soybean paste stew are classified as sodium-rich foods/salty foods, it is necessary to consider that they are fermented foods and the beneficial effects of cooking them with other vegetables. We have described these food items further in the Discussion section (Lines 251–253).

  1. For the purpose of transparency, please briefly state whether statistical analysis model assumptions were checked and validated.

→ Authors: We described the process of confirming and verifying the assumptions for the statistical analysis model in this study, and a related sentence was added to the Statistical Analysis in the Method section (Lines 130–136).

Reviewer 2 Report

This is an interesting work in which the author propose the use of a modified recommended food score for the identification of people's adherence to the DASH diet. I have several minor comments:

  • Exercise is an important factor influencing hypertension risk. Could the authors elaborate on how the categorization of performing regular exercise or not was made?
  • In section 2.4 it is not clear whether the quantity of consumption is reflected in the scoring system. The explanation does not seem to indicate it. Is the mRFS solely based in consumption frequency?
  • Also in section 2.4, the authors need to justify on which basis did they include specific foods as sodium-rich foods (e.g. due to having more than X g Na per 100 g food).
  • I understand the quintiles were generated from the total population, and not sex-stratified. Could the authors especify and explain why they did it this way?
  • Table 2 needs to be improved in the legend according to the statistical analyses made to obtain the indicated P-trend. Also, pairwise differences should be indicated in the individual values of each row. In this table, there is a possible mistake in the values of Q5 belonging to "Vegetables, score out of 15", since values are around 19 in the total sample, males and females. 
  • Table 3 needs further explanation of the statistical methods applied in the legend and, more importantly, needs to be revised since it is a copy of Table 4 and the data shown are not the means and standard deviations of SBP and DBP.
  • The legend of Table 4 also needs a bit further explanation of the statistical method applied.
  • It would be interesting to include a paragraph in the discussion concerning the lower significance of model 2 results, and perhaps also to especify which confounder is the most probable cause for significance to decrease. The complete results of the model should be able to bring some light to this issue. In general, the discussion needs a bit of improvement and further explanation of the findings.
  • Also, the results/discussion could benefit from the inclusion of observed preliminar cutoffs of this mRFS index, where it to be used by other researchers. 

Author Response

Reviewer 2

This is an interesting work in which the author proposes the use of a modified recommended food score for the identification of people's adherence to the DASH diet. I have several minor comments:

→ Authors: Thank you for these valuable comments.

Exercise is an important factor influencing hypertension risk. Could the authors elaborate on how the categorization of performing regular exercise or not was made?

→ Authors: Thank you for these valuable comments. Accordingly, we clarified whether regular exercise was performed by the subjects (Lines 87–96).

In section 2.4 it is not clear whether the quantity of consumption is reflected in the scoring system.

The explanation does not seem to indicate it. Is the mRFS solely based in consumption frequency?

→ Authors: The mRFS is scored by considering only the frequency of consumption without considering the quantity of consumption. The previously developed RFS was also a tool that considered only the frequency of consumption of food by investigating whether or not to eat the food at least once a week. In this way, the RFS can be used as a simple tool to evaluate the quality of the overall diet (Line 107).

Also, in section 2.4, the authors need to justify on which basis did they include specific foods as sodium-rich foods (e.g. due to having more than X g Na per 100 g food).

→ Authors: Thank you for these critical comments. We further described in section 2.4 the criteria for the inclusion of sodium-rich foods (Lines 117–121).

I understand the quintiles were generated from the total population, and not sex-stratified. Could the authors specify and explain why they did it this way?

→ Authors: We divided subjects into quintiles according to their mRFS by gender. This is clearly stated (Line 132).

Table 2 needs to be improved in the legend according to the statistical analyses made to obtain the indicated P-trend. Also, pairwise differences should be indicated in the individual values of each row. In this table, there is a possible mistake in the values of Q5 belonging to "Vegetables, score out of 15", since values are around 19 in the total sample, males and females.

→ Authors: The mRFS is a score with continuous values. However, in this study, we divided the mRFS into quintile groups. In this way, variables that were originally continuous are classified into tertiles, quartiles, or quintiles according to their distribution in many studies. When comparing the mean of certain variables by this group, statistical significance is often shown by the p-trend [e–g]. In other words, the analysis is performed under the assumption that there is a linear relationship between mRFS and a certain variable (factor). For this reason, we did not modify the original p-trend results in Table 2.

[e] Eng, J.Y., Moy, F.M., Bulgiba, A., Rampal, S Dose–response relationship between western diet and being overweight among teachers in Malaysia. Nutrients 2020, 12, 3092.

[f] Joo, N.-S., Dawson‐Hughes, B., Kim, Y-.S., Oh, K., Yeum, K.-J. Impact of calcium and vitamin d insufficiencies on serum parathyroid hormone and bone mineral density: analysis of the Fourth and Fifth Korea National Health and Nutrition Examination Survey (KNHANES IV-3, 2009 and KNHANES V-1, 2010). JBMR 2013, 28, 764–770.

[g] Park, S.-H., Lee, K.-S., Park, H.-Y. Dietary carbohydrate intake is associated with cardiovascular disease risk in Korean: Analysis of the third Korea National Health and Nutrition Examination Survey (KNHANES III). Int. J. Cardiol. 2010, 139, 234–240.

Table 3 needs further explanation of the statistical methods applied in the legend and, more importantly, needs to be revised since it is a copy of Table 4 and the data shown are not the means and standard deviations of SBP and DBP.

→ Authors: We inadvertently copied the results from Table 4 into Table 3. We apologize for this oversight. Table 3 has been properly re-edited (Table 3). We also added a further explanation of the applied statistical methods in the legend.

The legend of Table 4 also needs a bit further explanation of the statistical method applied.

→ Authors: We added a further explanation of the applied statistical methods in the legend of Table 4.

It would be interesting to include a paragraph in the discussion concerning the lower significance of model 2 results, and perhaps also to specify which confounder is the most probable cause for significance to decrease. The complete results of the model should be able to bring some light to this issue. In general, the discussion needs a bit of improvement and further explanation of the findings.

→ Authors: Thank you for the valuable comment. The significance of Model 2 decreased in the total participants and women. Among the variables additionally adjusted in Model 2, energy intake was found to be significantly related to the risk of high blood pressure in men, but not in women. Therefore, it is thought that energy intake between men and women may not have acted as such a cause (the significance of Model 2 decreased in the total participants and women). We reinforced the Discussion section a little (Lines 206, 212–218, 231–233, 251–253, 259–262, 271–272).

Also, the results/discussion could benefit from the inclusion of observed preliminar cutoffs of this mRFS index, where it to be used by other researchers.

→ Authors: As described in the Discussion section, some areas require confirmation of the results of this study, so it is burdensome to emphasize the cutoff values, and hence, no additional descriptions are made in the Discussion section. However, the mRFS scores for each quintile are also presented in the table to show the corresponding cutoff values.

Round 2

Reviewer 2 Report

Authors have addressed the comments successfully.